# Recurrent Cystitis in Women—A Real-World Analysis of Bacteria Spectrum and Resistance Situation for Calculated Therapy

**DOI:** 10.3390/antibiotics13090890

**Published:** 2024-09-16

**Authors:** Philipp J. Spachmann, Maximilian Radlmaier, Stefan Denzinger, Maximilian Burger, Johannes Breyer, Wolfgang Otto, Marco J. Schnabel, Daniel Vergho

**Affiliations:** 1Department of Urology, Caritas St. Josef Medical Center Regensburg, University of Regensburg, 93053 Regensburg, Germany; 2Urologie Landau and Urologie Plattling, 94405 Landau/Isar, Germany

**Keywords:** urinary tract infection, recurrent cystitis, antibiotic resistance, antibiotic stewardship, calculated antibiosis

## Abstract

Recurrent cystitis in women represents an everyday challenge; however, little to no data regarding this population are available. This study aimed to evaluate this collective with respect to a rational calculated antibiotic therapy. Urine cultures and antibiograms from a urological office were retrospectively evaluated from patient data collected between January 2017 and June 2019. The evaluation was conducted using SPSS ©. In total, 84 female patients, who were aged between 18 and 87 years old (median 60 years), suffered from recurrent cystitis. *Escherichia coli* was found in 53.9% of cases, *Staphylococcus aureus* and enterococci were each found in 6.7%, and *Proteus* spp. and *Streptococcus agalactiae* were each found in 5.6%. The resistance levels to ciprofloxacin (CIP), trimethoprim–sulfamethoxazole (TRS), nitrofurantoin (NIT), and nitroxoline (NOX) were 18.2%, 30.7%, 16.1%, and 12.5% in the tested cases, respectively. Regarding *E. coli*, resistance to CIP, TRS, and NIT was found in 17.8%, 25%, and 4.2% of the tested cases, and no resistance to NOX was found. The resistance level to CIP was in a tolerable range of <20% in the overall cohort and the *E. coli* subgroup. More than a quarter of the bacteria were resistant to TRS. The low resistance rates for NIT and NOX are remarkable, promoting the use of these substances if they are not yet used.

## 1. Introduction

Recurrent cystitis in women represents an everyday challenge in infectious urology and gynecology. The European Association of Urology (EAU) defines recurrent cystitis as three or more episodes in 12 months or two or more episodes in 6 months [1]. The affected patients are often severely stressed, as well as psychologically affected, due to the lack of long-term therapeutic success and frequent antibiotic therapies. Moreover, the quality of life of patients is usually reported to be limited. Complicating and predisposing factors in post-menopausal women include urinary incontinence, cystocele potentially associated with increased residual urine after miction, and vaginal hormone deficiency with local dysbiosis [2]. Sexual intercourse, particularly with new or changing sexual partners, is a further risk factor, especially for pre-menopausal women [1]. As recurrent cystitis is one of the most common bacterial infections, this topic is of socio-economic importance [3,4,5,6].

The pathogen spectrum and resistance situation in uncomplicated cystitis have been identified [7,8,9]; however, little to no clinical data are available on the spectrum of bacteria and resistance levels for recurrent cystitis in women. The German S3 guideline on urinary tract infections (UTI) does not include recommendations on this topic, only discussing uncomplicated urinary tract infections such as uncomplicated cystitis [10]. The European guideline focuses on the prophylaxis of recurrent cystitis and risk factors for pre- and post-menopausal women but lacks a recommendation for treatment in an acute episode of recurrence [1]. Furthermore, the Canadian guideline from 2011 is limited regarding its definition, risk factors, and prophylaxis [11]. Immediate therapy is often strongly needed in the situation of recurrence, due to the extent of the symptoms, requiring a calculated antibiotic therapeutic strategy. This study aimed to evaluate a cohort of women with recurrent cystitis, specifically regarding a rational and pragmatic calculated antibiotic therapy in an acute episode of recurrence.

## 2. Results

Of all cases, 361 (59.6%) were appearances of women with complaints. Recurrent cystitis was found in 83 cases (23% of women with complaints). The patients were between 18 and 87 years old (median: 60 years). In total, 20 patients (24.7%) belonged to the pre- and 63 (75.3%) to the post-menopausal cohorts. The pre-menopausal cohort was between 18 and 49 years old (median: 28 years). The post-menopausal group comprising 63 patients was between 50 and 87 years old (median: 65 years).

*Escherichia coli* was found in 57% of cases (n = 48), *Staphylococcus aureus* and enterococci in 7.1% each (n = 6), and *Proteus* spp. and *Streptococcus agalactiae* in 6% each (n = 5). The proportion of the other bacteria was <5% (Table 1).

The overall resistance levels to ciprofloxacin, trimethoprim–sulfamethoxazole, nitrofurantoin, and nitroxoline were 18.9%, 29.3%, 17.3%, and 12.3% in the tested cases, respectively (Table 2).

Regarding the most common bacteria, *E. coli*, resistance to ciprofloxacin, trimethoprime-sulfamethoxazole, and nitrofurantoin was found in 17.8%, 25%, and 4.2% of the tested cases, respectively. Susceptibility to nitroxoline was evident in all tested cases (Table 3).

In the pre-menopausal women (Table 4), *E. coli* was evident in 12 urine culture cases (60%), and *Proteus* spp. and Strept. agalactiae were found in 2 urine cultures each (10%). *S. aureus*, *Klebsiella oxytoca*, *Serratia marcescens*, and enterococci were each found in one culture (5% each).

Of the tested bacteria, 18.8% (3 out of 16 cases) were resistant to ciprofloxacin, 26.3% to trimethoprim–sulfamethoxazole (5 out of 19 cases), and 11.1% to nitrofurantoin (2 out of 18 cases). Only 1 urine culture of the 16 tested was resistant to nitroxoline (6.3%) (Table 2). The analysis for *E. coli* was not rational, due to the small number of only 12 findings in pre-menopausal women, but it can be found in Table 3.

In post-menopausal women, the spectrum of bacteria (Table 5) indicated *E. coli* in 36 cultures (57.1%) and *S. aureus* and enterococci in 5 each (7.9%). Other bacteria were found in less than 5% of the cultures. Resistance to ciprofloxacin, trimethoprim–sulfamethoxazole, nitrofurantoin, and nitroxoline was present in 19% (11 cases), 30.2% (19 cases), 19% (12 cases), and 14.5% (8 findings), respectively (Table 2). *E. coli* was resistant to ciprofloxacin, trimethoprim–sulfamethoxazole, and nitrofurantoin in the post-menopausal women in 14.7% (5 findings), 25% (9 findings out of 36 tests), and 5.6% (2 findings), respectively. No resistance to nitroxoline was found in the 36 tested cases (Table 3).

## 3. Discussion

The treatment of recurrent cystitis is becoming more complex, as most patients present without any previous microbiological findings, and given that no culture has been previously performed. This follows the German guidelines for general medicine, as no culture must be performed in the first episode of cystitis [10].

The microbiomes of the intestines and the urinary bladder can also influence the risk of developing recurrent cystitis and are simultaneously negatively influenced by frequent antibiotic therapies [12]. The hormonal changes in women during menopause can also lead to changes in the vaginal microbiome and bladder, with a subsequent increased susceptibility to recurrent cystitis [13].

Significantly, symptoms of cystitis such as abdominal pain, pollakiuria, urgency, and dysuria can also derive from other clinical pictures and illnesses (e.g., cystocele), which can promote urinary tract infections, cystitis, and, subsequently, recurrent cystitis.

Theoretically, asymptomatic bacteriuria can be present alongside these non-infectious causes of these symptoms, which further complicates the distinction.

On one hand, this underlines the need for an accurate and detailed anamnesis, and, on the other hand, it highlights the need for good knowledge of the affected patient cohort, including the bacterial spectrum and resistance rates [6,14].

The development of resistance against antibiotics due to the frequent use of antibiotic substances with the subsequent development of multi-resistant bacteria can also occur with recurrent cystitis [15]. Antibiotic stewardship strategies are highly relevant against this background [16]. The patient’s adherence to therapy is an important risk factor for developing recurrent cystitis. Data show that only two-thirds of patients are compliant with antibiotic therapy [17]. Moreover, taking antibiotics inconsistently increases the risk of resistance-development [18]. The most important topics in recurrent cystitis are the identification of risk factors and the prophylaxis of recurrent cystitis [14].

In our cohort, 23% of females were symptomatic with recurrent cystitis, which is higher than previously reported [10,19,20,21,22,23]. A plausible reason might be the pre-selection of the collective, considering that the patients already presented with an initial urinary tract infection in urological treatment, while patients treated by a general practitioner or gynecologist were not recorded.

In contrast with the literature [1,10,24,25], the rate of recurring infections in the investigated collective with complaints was not significantly lower in the group of pre-menopausal women, with 26.5%, than in the group of post-menopausal women, with 24.1% (*p* = 0.665). This contrast with the existing literature might have occurred due to the pre-selection of patients but remains remarkable. 

*E. coli* was the main bacteria found, following the literature on acute uncomplicated cystitis in women in Germany [9,25]. The frequency of evidence of unspecific enterococci was remarkable. Here, likewise, only commensal colonization must be considered. *E. coli* in the total cohort showed comparable susceptibility to the results of the ARESC study on acute uncomplicated cystitis in women regarding nitrofurantoin (95.8 vs. 95.4% ARESC) and trimethoprim–sulfamethoxazole (75% vs. 74% ARESC). Regarding ciprofloxacin, the susceptibility rates were lower in this cohort (82.2 vs. 95.4% ARESC) [9]. Nitroxoline was not tested in the ARESC study. In a Japanese nationwide study on acute uncomplicated cystitis in women, the susceptibility of *E coli* in the total cohort was comparable between ciprofloxacin (82.2 vs. 80.2%) and nitrofurantoin (95.8 vs. 89.6%), with a trend of higher susceptibility in this study [26]. Trimethoprim–sulfamethoxazole and nitroxoline were not tested in this study.

The resistance level to ciprofloxacin was in a tolerable range and, compared with other findings, in a sound range in the total collective and for the main bacteria *E. coli*. However, its use should be restricted due to the existing side-effect profile and the importance of fluoroquinolones as a substance class in the treatment of severe systemic—not only urological—diseases, although approval in the situation of a complicated urinary tract infection may be justified. The high resistance rate to trimethoprim–sulfamethoxazole of more than 20% might be related to its frequent use as a first-line treatment in uncomplicated acute cystitis. A calculated treatment of a symptomatic recurrence of cystitis with trimethoprim–sulfamethoxazole cannot be advised, according to the examined collective. The low resistance rates to nitrofurantoin and nitroxoline in the examined collective were noteworthy; these substances represent a reasonable first choice for calculated antibiotic therapy in recurring infections. This is surprising in recurrent cystitis, as these substances are also the first choice in guidelines for uncomplicated urinary tract infections. However, this might be explained by the initial choice of treatment in the first episode of uncomplicated acute cystitis, which may have caused the extremely low resistance rates. Considering that nitrofurantoin and nitroxoline have been used in clinical practice since the 1950s and 1960s, respectively (often in Germany), our findings further underline the importance of these two agents due to the low observed resistance rates of *E. coli* to nitrofurantoin, in agreement with national German data [27].

Overall, this study provides an up-to-date description of a collective of women with recurrent cystitis regarding the range of bacteria, the resistance situation, and their antibiotic resistance pattern in a urological office. The results concerning the bacterial spectrum, resistance level, and respective susceptibilities of the bacteria are comparable in many aspects to data from the literature regarding acute uncomplicated cystitis in women.

## 4. Materials and Methods

Between 01/2017 and 06/2019, 800 consecutive voided midstream urine cultures and antibiograms of males and females from a urological office were retrospectively reviewed and evaluated in the context of the patients’ clinical data and sex. Male patients were excluded. The definition of recurrent cystitis was based on the EAU’s definition specifying three or more episodes in 12 months and two or more episodes in 6 months. The group of pre-menopausal women was defined as being aged below 50 years, according to the mean ages at menopause found in the literature [28,29,30]. Urine cultures were obtained in every appearance with complaints.

The urine was voided midstream urine. Patient information on the technique and disinfection was given before. The microbiological cultivation and evaluation took place in the office of urology and were conducted by the team and were performed according to the recommendations in Ivo Beyaert’s guidelines for microbiology for urologists [31].

An amount of 100 microliters of urine was inoculated using a disposable pipette onto plates containing cystine–lactose–electrolyte-deficient (CLED) agar, MacConkey agar, blood Columbia colistin (polymyxin E) and nalidixic acid (CNA) agar, and Sabouraud glucose agar.

The inoculated plates were incubated overnight in an incubator at 36 ± 1 °C. The incubated plates were removed from the incubator the following day and checked. Schemes for optical comparison regarding estimation of the number of colony-forming units (CFUs) were available.

The plates were checked as follows: 

CLED agar: Yellow or yellowish discoloration of the agar due to CFUs meant that the bacterium utilized lactose. The absence of discoloration when CFUs were detected, or bluish discoloration, meant that a lactose-negative bacterium was present.

MacConkey agar: Red or reddish CFUs were present in lactose-positive bacteria. Any inherent color of the bacterial colonies needed to be considered and compared with the result of the CLED agar reaction regarding the lactose reaction. Colorless colonies on unchanged red–brown agar meant that no lactose utilization occurred. Some bacteria could also change the color of the agar to ocher-yellow if they did not utilize lactose. In addition, testing for an oxidase reaction was performed. For this purpose, a CFU, with the eyelet, was painted onto the oxidase indicator paper. If the oxidase reaction was positive, a dark blue discoloration occurred. Here, the bacteria’s own colors could cause false-positive reactions. In case of doubt, a CFU could be taken from the CLED agar. In case of a positive reaction, it was considered as *Pseudomonas* spp., no further testing was conducted, and an antibiogram was created. If the reaction was negative, a so-called EnteroPluri test was performed to identify Gram-negative, oxidase-negative bacteria. For this purpose, another CFU was removed, introduced into the test system, and incubated in the incubator at 36 ± 1 °C for at least 18 h.

Blood–CNA agar: This test indicated the hemolysis of the agar by the bacterium. A positive reaction occurred if it turned green or greenish. In the case of streptococci, the degree of hemolysis can provide information about the type: α in greening, with incomplete hemolysis; β in complete hemolysis; and γ without hemolysis. Yeast and other fungi can also grow on this agar. Therefore, a cross-check needed to be made, and any growth on the Sabouraud agar needed to also be considered. Additionally, multi-resistant Gram-negative bacteria, such as *Pseudomonas* spp., *Proteus* spp., and *Klebsiella* spp., can also grow on this agar. The frequency is rare, but it has become increasingly apparent due to the increase in resistance.

To evaluate the catalase reaction, a CFU was transferred to a slide and then mixed with the reagent. If there was an immediate “bubbling” reaction, a positive reaction occurred, and the bacteria tested were staphylococci, which were further differentiated with the TetraStaph test. If the result was negative, further workup was conducted using an enterococcal plate. For the TetraStaph, two CFUs were inserted into the test system with an eyelet. The test system was then incubated for at least 18 h at 36 plus/minus 1 °C. To create the enterococci plate, a CFU was applied to it using a cotton swab and spread. The test system was also incubated for at least 18 h at 36 ± 1 °C.

Sabouraud agar: Only yeasts and other fungi grow on this agar. No further testing was conducted, but the result was documented for comparison.

For the antibiogram, a Müller–Hinton plate was used, distinguishing between the antibiograms for Gram-positive and Gram-negative bacteria. Four CFUs were removed from the corresponding agar and mixed with NaCl. A dilution smear was conducted using McFarland medium. The liquid was then spread on the Müller–Hinton plate with a cotton swab and incubated again for at least 18 h at 36 ± 1 °C.

The bacterial species and strains were evaluated by comparing the results of the incubated plates, according to the respective codes for the EnteroPluri test or flow charts for the TetraStaph. Enterococci were detected when the enterococcal plate turned black. A negative result was given for staphylococci for all tests.

Antibiograms were made by applying an appropriate inhibition zone film to the plates for each bacterium. The results of the reaction to the respective antibiotic substances were read, compared, and recorded regarding susceptibility, susceptibility to high doses, and resistance.

The following antibiotics were included in the investigation. Ciprofloxacin must not be used in uncomplicated urinary tract infections, as recommended by the European Medicines Agency (EMA). However, in complicated UTIs such as recurrent cystitis, trimethoprim–sulfamethoxazole is a traditionally frequently used agent in cystitis treatment. Nitroxoline and nitrofurantoin were included as two other orally available substances. First-line agents in the treatment of acute uncomplicated cystitis, such as fosfomycin and pivmecillinam, were not investigated due to their limited approval for uncomplicated cystitis only. Cephalosporines were not analyzed due to their limited oral bioavailability.

Previous urine cultures could not be found in most cases, as those patients were seen by a general practitioner without collecting a urine culture [32].

Cases with missing parameters and patients under the age of 18 were excluded. The evaluation was performed using SPSS ©.

There are several strengths and limitations of this study:

This study is limited due to its retrospective, unicentric local character and the small number of cases and included substances, due to the examination panel in the executing urological office. Nevertheless, it provides insights into an under-investigated field. 

The age of menopause was defined as 50 years or more, and the exact status of menopause was not asked. This study was performed in Germany, which is not a limitation but is important due to the occurrence of regional differences.

A strength of this study is that it provides an up-to-date description of a collective of women with recurrent cystitis in the relevant fields of urology, general medicine, and gynecology.

## 5. Conclusions

Based on the considered data, this study provided an up-to-date recommendation for the treatment of recurrent cystitis in women. This is important, despite its retrospective character and the small number of patients, due to its unicentric local character, as there is currently little to no evidence regarding this frequent female illness.

The rare side effects, the cost-efficiency, and the possibility of use of nitrofurantoin and nitroxoline as long-term prophylaxis represent further benefits.

The accurate and rapid identification of risk factors, their elimination (if possible), and the development of existing respective evaluations of non-antibiotic strategies remain necessary.

## Figures and Tables

**Table 1 antibiotics-13-00890-t001:** Bacteria spectrum of women with recurrent cystitis (n = 83).

Bacterium	n	%
*E. coli*	48	57
*S. aureus*	6	7.1
enterococci	6	7.1
*Proteus* spp.	5	6
*Strept. agalactiae*	5	6
*Enterobacter cloacae*	3	3.6
*Klebsiella oxytoca*	3	3.6
*Pseudomonas aeruginosa*	3	3.6
*Klebsiella pneumoniae*	2	2.4
*Citrobacter species*	1	1.2
*Serratia marcescens*	1	1.2

**Table 2 antibiotics-13-00890-t002:** Antibiotic resistance situation of women, pre-menopausal women, and post-menopausal women with recurrent cystitis.

Antibiotics	Tested n (All)	Resistance % (All)	Tested n (Pre-Menopausal)	Resistance % (Pre-Menopausal)	Tested n (Post-Menopausal)	Resistance % (Post-Menopausal)
Ciprofloxacin	74	18.9	16	18.8	58	19
Trimethoprime-sulfamethoxazole	82	29.3	19	26.3	63	30.2
Nitrofurantoin	81	17.3	18	11.1	63	19
Nitroxoline	71	12.7	16	6.3	55	14.5

**Table 3 antibiotics-13-00890-t003:** Antibiotic resistance situation of *E. coli* in women, pre-menopausal women, and post-menopausal women with recurrent cystitis.

Antibiotics	Tested n (All)	Resistance % (All)	Tested n (Pre-Menopausal)	Resistance % (Pre-Menopausal)	Tested n (Post-Menopausal)	Resistance % (Post-Menopausal)
Ciprofloxacin	45	17.8	11	27.3	34	14.7
Trimethoprime-sulfamethoxazole	48	25	12	25	36	25
Nitrofurantoin	48	4.2	12	0	36	5.6
Nitroxoline	47	0	11	0	36	0

**Table 4 antibiotics-13-00890-t004:** Bacteria spectrum of pre-menopausal women with recurrent cystitis (n = 20).

Bacterium	n	%
*Escherichia coli*	12	60
*Proteus* spp.	2	10
*Strept. agalactiae*	2	10
*S. aureus*	1	5
*Klebsiella oxytoca*	1	5
*Serratia marcescens*	1	5
enterococci	1	5

**Table 5 antibiotics-13-00890-t005:** Bacteria spectrum of post-menopausal women with recurrent cystitis (n = 63).

Bacterium	n	%
*Escherichia coli*	36	57.1
*Staphylococcus aureus*	5	7.9
enterococci	5	7.9
*Enterobacter cloacae*	3	4.8
*Proteus* spp.	3	4.8
*Streptococcus agalacticae*	3	4.8
*Pseudomonas aeruginosa*	3	4.8
*Klebsiella pneumoniae*	2	3.2
*Klebsiella oxytoca*	2	3.2
*Citrobacter species*	1	1.6

## Data Availability

All data generated during this study can be enquired from the corresponding author.

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
