# Peer review of "Recurrent Cystitis in Women—A Real-World Analysis of Bacteria Spectrum and Resistance Situation for Calculated Therapy"

_antibiotics, 2024, doi:10.3390/antibiotics13090890_

Round 1
Reviewer 1 Report
Comments and Suggestions for Authors
The study by Spachmann and his colleagues provide updated data on antibiotic resistance in a cohort of patient switch recurrent UTI. The data is easy to follow and the discussion is sound and bring up the major limitation of the study. I only have minor concerns with the current manuscript.
in the introduction the authors talk about phototherapy and never talks about this or other alternative treatments again.
1. Either remove this paragraph or discuss this in relation to the findings in the discussion.
2. there are obviously other alternative treatments including immunomodulation and vaccine strategies, if you keep the paragraph these strategies should be mentioned as well.
Comments on the Quality of English LanguageThe authors consistently sends paragraphs when the paragraph is obviously not finished l.
Author Response
Comment 1: in the introduction the authors talk about phototherapy and never talks about this or other alternative treatments again.
1. Either remove this paragraph or discuss this in relation to the findings in the discussion.
2. there are obviously other alternative treatments including immunomodulation and vaccine strategies, if you keep the paragraph these strategies should be mentioned as well.
Thank you, we agree with you and removed the paraphraph.
Comment 2: The authors consistently sends paragraphs when the paragraph is obviously not finished l.
Thank you fo that comment, please see the canges in the manuscript.

Reviewer 2 Report
Comments and Suggestions for Authors
Throughout the document, the names of the genus and species of bacteria should be italicized.
Abbreviations of antimicrobial drugs please take from the official EUCAST document - https://view.officeapps.live.com/op/view.aspx?src=https%3A%2F%2Fwww.eucast.org%2Ffileadmin%2Fsrc%2Fmedia%2FPDFs%2FEUCAST_files%2FGuidance_documents%2FAntimicrobial_abbreviations_v7_20220120.xlsx&wdOrigin=BROWSELINK
Line 11 – abbreviations "(UC)" and "(AB)" are unnecessary in the abstract
Line 15 – "Enterococci" should be written in lower case, it is a common name, please correct it in the whole document
Line 15 – instead of "Proteus species" it should be abbreviated as "Proteus spp.", the word “Proteus” in italics, and “spp.” not
Line 16 – please do not use the name “cotrimoxazole” it is a nasty colloquialism, the proper names of both substances should be used, i.e. “trimethoprim/sulfamethoxazole”
Line 40 – missing source (citation)
Line 42 – “intestines” instead of “intestine”
Line 53 – missing source (citation)
Line 56 – missing source (citation)
Lines 64-67 – missing source (citation)
In my opinion, the Introduction is written too generally. The authors should rewrite these sections, developing each of the issues raised in 2-3 sentences. In the introduction, the authors should rely exclusively on current, published knowledge, which is why each paragraph/fragment should have a citation source indicated. There is room for your own considerations in the Discussion section.
Line 72 – the first sentence of the paragraph is quite confusing
Table 1 – Weren’t Staph. epidermidis and Micrococcus spp. contaminations?
Table 2 and 3 – what ‘Antibiosis’ mean?
All tables – the table titles all need to be rewritten. In their current form they look like a jumble of random words. This is definitely not correct English.
Line 94 – please don’t use “sensitivity” but “susceptiblity”, correct it in the whole document, this is about the activity of drugs against bacteria, not their feelings
Line 100, 101 – do not use the abbreviation “UC” at all
Lines 124-125 – age of menopause should be described in Methods section, not in Discussion
Lines 131-136 – This paragraph should be moved to a new subsection “Strength and Limitations”.
Lines 142-155 – this should be in Introduction
Line 161 – add source, why resistance lower than 20% is tolerable
Generally, the Discussion should be expanded with more comparisons with other current data. Furthermore, a comparison with the etiology of other forms of UTI, such as lower or upper UTI or CAI and HAI infection, could be added. The authors should develop this section properly.
Line 186 – abbreviations “(UC)” and “(AB)” are unnecessary
Lines 203-204 – paragraph unnecessary in this section, move to Discussion
Generally, Materials and methods are selected correctly, but their description should be expanded. In which unit were the patients treated, whose urine samples were analyzed? In what process did the microbiology laboratory process the samples? Which antibiotics were marked in the antibiogram? By what methods? On what basis did the laboratory reject the material for testing? Did the patients themselves pass urine for testing? By what method? How was contamination of the material avoided? This is just the tip of the iceberg of questions that can be asked to this section. Also, a subsection called "Strength and Limitations" should be added.
The Conclusions section is okay in terms of content, but the English needs to be improved, because it is difficult to read and understand.
Comments on the Quality of English Language
The English of the article is poor, it needs thorough and extensive proofreading by a native speaker.
Author Response
Comment 1: Throughout the document, the names of the genus and species of bacteria should be italicized. (...)
Thank you, please see the changes in the manuscript.
Comment 2: Line 11 – abbreviations "(UC)" and "(AB)" are unnecessary in the abstract.
Thank you, please see the changes in the manuscript.
Comment 3: Line 15 – "Enterococci" should be written in lower case, it is a common name, please correct it in the whole document
Thank you, please see the changes in the manuscript.
Comment 4: Line 15 – instead of "Proteus species" it should be abbreviated as "Proteus spp.", the word “Proteus” in italics, and “spp.”
Thank you, please see the changes in the manuscript.
Comment 5: Line 16 – please do not use the name “cotrimoxazole” it is a nasty colloquialism, the proper names of both substances should be used, i.e. “trimethoprim/sulfamethoxazole”
Thank you, please see the changes in the manuscript.
Comment 6: Line 40 – missing source (citation)
Thank you, please see the changes in the manuscript.
Comment 7: Line 42 – “intestines” instead of “intestine”
Thank you, please see the changes in the manuscript.
Comment 8: Line 53 – missing source (citation)
Thank you, please see the changes in the manuscript.
Comment 9: Line 56 – missing source (citation)
Thank you, please see the changes in the manuscript.
Comment 10: Lines 64-67 – missing source (citation)
Thank you, please see the changes in the manuscript.
Comment 11: In my opinion, the Introduction is written too generally. The authors should rewrite these sections, developing each of the issues raised in 2-3 sentences. In the introduction, the authors should rely exclusively on current, published knowledge, which is why each paragraph/fragment should have a citation source indicated. There is room for your own considerations in the Discussion section.
Thank you, please see the changes in the manuscript.
Comment 12: Line 72 – the first sentence of the paragraph is quite confusing
Thank you, please see the changes in the manuscript.
Comment 13: Table 1 – Weren’t Staph. epidermidis and Micrococcus spp. contaminations?
Thank you, please see the changes in the manuscript, the statisctis were recalculated without these bacteria.
Comment 14: Table 2 and 3 – what ‘Antibiosis’ mean?
Thank you, please see the changes in the manuscript, antibiotic substances were meant
Comment 15: All tables – the table titles all need to be rewritten. In their current form they look like a jumble of random words. This is definitely not correct English.
Thank you, please see the changes in the tables.
Comment 16: Line 94 – please don’t use “sensitivity” but “susceptiblity”, correct it in the whole document, this is about the activity of drugs against bacteria, not their feelings
Thank you, please see the changes in the manuscript
Comment 17: Line 100, 101 – do not use the abbreviation “UC” at all
Thank you, please see the changes in the manuscript
Comment 18: Lines 124-125 – age of menopause should be described in Methods section, not in Discussion
Thank you, please see the changes in the manuscript
Comment 19: Lines 131-136 – This paragraph should be moved to a new subsection “Strength and Limitations”.
Thank you, please see the changes in the manuscript
Comment 20:Lines 142-155 – this should be in Introduction
Thank you, please see the changes in the manuscript
Comment 21: Line 161 – add source, why resistance lower than 20% is tolerable
Thank you, please see the changes in the manuscript, the passage regarding 20% was removed.
Comment 22: Generally, the Discussion should be expanded with more comparisons with other current data. Furthermore, a comparison with the etiology of other forms of UTI, such as lower or upper UTI or CAI and HAI infection, could be added. The authors should develop this section properly.
Comment 23: Line 186 – abbreviations “(UC)” and “(AB)” are unnecessary
Thank you, please see the changes in the manuscript.
Comment 24: Lines 203-204 – paragraph unnecessary in this section, move to Discussion
Thank you, please see the changes in the manuscript.
Comment 25: Generally, Materials and methods are selected correctly, but their description should be expanded. In which unit were the patients treated, whose urine samples were analyzed? In what process did the microbiology laboratory process the samples? Which antibiotics were marked in the antibiogram? By what methods? On what basis did the laboratory reject the material for testing? Did the patients themselves pass urine for testing? By what method? How was contamination of the material avoided? This is just the tip of the iceberg of questions that can be asked to this section. Also, a subsection called "Strength and Limitations" should be added.
Thank you, please see the changes in the manuscript
The Conclusions section is okay in terms of content, but the English needs to be improved, because it is difficult to read and understand.
Thank you, please see the changes in the manuscript
Round 2
Reviewer 2 Report
Comments and Suggestions for Authors
See the attachment.

Minor editing of English language required.
Author Response
Most of the comments were taken into account and the article gained in quality. The corrections
were made chaotically and a bit sloppily. Without an editing mode, so I had to go through the entire
text again, looking for what and where was (or was not) corrected. There are still a few important
issues that need to be corrected to make it suitable for publication:
Answer: First of all we would like to apologize for the effort required, the proof reading and subsequent corrections made it very confusing, as the text as a whole was revised intensively.
• bacterial species names are not italic, this should be corrected throughout the manuscript!!!
Answer: Thank you, we agree, please see the highlighted changes in the manuscript
• when the name of a bacterial species appears for the second or subsequent time in the text proper (excluding the abstract), the generic name can be abbreviated, e.g. S. agalactiae, S. aureus
Answer: Thank you, please see the highlighted changes in the manuscript
• “cotrimoxazole” has not yet been replaced everywhere with “trimethoprim-sulfamethoxazole” (lines 155, 157) – check the whole article for this
Answer: Thank you, please see the highlighted changes in the manuscript
• in the abstract (lines 18, 21) the abbreviation “COT” remains, which needs to be changed to “TRS” - check the whole article for this
Answer: Thank you, please see the highlighted changes in the abstract in these lines
• in the tables there is still the word "Antibiosis", which should be replaced with "Antibiotics" or "Antimicrobials"
Answer: Please see the highlighted changes in the tables
• lines 15 & 79 - remove “UC” abbreviation
Answer: Please see the higlighted changes in the manuscript in these lines
• use the abbreviations that you have introduced in the text consistently – this is especially true for the names of antimicrobials, you cannot write the abbreviation once and the full name another time
Answer: We agree, please see the higlighted changes in the manuscript
• Tables 1, 4, 5 - "enterococci" with lower case; "spp." instead of "species"
Answer: Please see the highlighted changes in these tables
• The table titles still do not describe what the table contains - I will not come up with suggestions for these titles, because I am not a co-author of the article; please look through the articles available in the publishers' databases to see how it should look
Answer: Please see the highlighted changes in the tables
• Line 213 - "CNA" not “CAN”
Answer: Please see the highlighted changes in the manuscript
The very comprehensive description of the methodology used deserves praise. Someone really worked hard.
Thank you very much.
The paragraphs of "Strength and Limitations" should be a subsection of "Materials and
Methods".
Answer: Please see the changes in the lines 260-270 in the section "Materials and Methods"
